# GeneToList: A Web Application to Assist with Gene Identifiers for the Non-Bioinformatics-Savvy Scientist

**DOI:** 10.3390/biology11081113

**Published:** 2022-07-26

**Authors:** Joshua D. Breidenbach, E. Francis Begue III, David J. Kennedy, Steven T. Haller

**Affiliations:** Department of Medicine, College of Medicine and Life Sciences, University of Toledo, Toledo, OH 43614, USA; eugene.f.begue@gmail.com (E.F.B.III); david.kennedy@utoledo.edu (D.J.K.); steven.haller@utoledo.edu (S.T.H.)

**Keywords:** gene nomenclature, web application, gene ID

## Abstract

**Simple Summary:**

With the increasing popularity of omics technologies, many scientists are dealing with large sets of genomic data for the first time. While there are many amazing bioinformatics tools available to help analyze this data, one common difficulty these scientists face is that not all gene data or analysis tools use the same genomic nomenclature. Another issue is that many publications still use obsolete or colloquial gene aliases instead of official gene identifiers. Common gene ID conversion tools and other downstream analysis software struggle with these gene aliases. Therefore, we developed a free and publicly available web application, GeneToList, to assist in gene ID disambiguation and gene ID conversion, with a specific focus on a user-friendly interface for the non-bioinformatics-savvy scientist.

**Abstract:**

The increasing incorporation of omics technologies into biomedical research and translational medicine presents challenges to end users of the large and complex datasets that are generated by these methods. A particular challenge in genomics is that the nomenclature for genes is not uniform between large genomic databases or between commonly used genetic analysis tools. Furthermore, outdated genomic nomenclature can still be found amongst scientific communications, including peer-reviewed manuscripts. Therefore, a web application (GeneToList) was developed to assist in gene ID conversion and alias matching, with a specific focus on achieving a user-friendly interface for the non-bioinformatics-savvy scientist. It currently includes gene information for over 38,000 different taxa retrieved from the National Center for Biotechnology and Information (NCBI) Gene resource. Supported databases of gene IDs include NCBI Gene Symbols, NCBI Gene IDs (Entrez IDs), OMIM IDs, HGNC IDs, Ensembl IDs, and 28 other taxa-specific identifiers. GeneToList is available at genetolist.com. The tool is a web application that is compatible with many standard browsers. The gene ID conversion feature of this application was found to outcompete the common gene ID conversion tools. Specifically, it was able to successfully convert all tested IDs, whereas the others were not able to recognize the gene aliases. Therefore, the gene ID disambiguation provided by this application should be beneficial for many scientists dealing with gene data when the uniformity of gene nomenclature is important for downstream analysis.

## 1. Introduction

The increasing popularity of omics technologies in biomedical research has led to the birth of a subfield of data science, bioinformatics. While these techniques are becoming crucial to research, it is important to recognize that not all who stand to benefit from these advancements are poised to learn programming languages or become bioinformaticians. Additionally, the myriad information attained through methods such as next-generation sequencing-based RNA-sequencing requires the community to constantly update the gene and protein nomenclature. When dealing with the complex datasets generated by these methods and attempting to utilize the many genetic analysis tools available, there is difficulty in matching the format of one output to the required input of another. Additionally, obsolete genomic nomenclature persists colloquially and amongst peer-reviewed manuscripts. While great efforts have been made to allow for the conversion of gene identifiers, these usually require advanced knowledge of programming languages (biomaRt, MyGene—https://mygene.info/ (accessed on 15 June 2022), and org.Hs.eg.db) [1,2]. Otherwise, there are a few web applications that provide a user interface for the conversion of gene IDs. However, some are intended as an initial step of a more complex and powerful tool instead of a dedicated application for this purpose (DAVID—https://david.ncifcrf.gov/home.jsp) [3]. Others are dedicated but rely on specific user input such as the input ID type and desired output, which may be a barrier for the unfamiliar scientist (g:Convert—https://biit.cs.ut.ee/gprofiler/ and bioDBnet—https://biodbnet-abcc.ncifcrf.gov/db/db2db.php) [4,5]. Importantly, we are not aware of any tool that assists in alias matching, especially in situations when obsolete IDs are ambiguous. Therefore, we set out to create a web application with a graphical user interface that can assist in the conversion of gene IDs and that disambiguates obsolete gene IDs in a high-throughput manner suitable for large lists of genes.

## 2. Materials and Methods

### 2.1. Data Collection

Gene information for more than 38,000 taxa were collected from the National Center for Biotechnology and Information (NCBI) Gene resource [6,7,8]. Therefore, the application supports any taxon with gene information stored by the NCBI, including archaea, fungi, invertebrates, mammalian and nonmammalian vertebrates, plants, protozoa, and viruses. Supported databases of gene IDs include NCBI Gene Symbols, NCBI Gene IDs (Entrez IDs), OMIM IDs, HGNC IDs, Ensembl IDs, and 28 more taxa-specific identifiers. A summary of the gene information is shown in Figure 1.

### 2.2. Application

This web application assists in 2 separate tasks. The first is disambiguating obsolete gene nomenclature. A single search term or a list of terms (separated by a comma or white space) can be entered into the input field (text box) and added to an existing list or used to start a new one. The search algorithm first starts by attempting to match searched terms as-is with a database of gene information for the selected taxonomy. Exact matches to official NCBI gene symbols are added directly to the Final List. Additionally, matches to the NCBI official gene symbols after only slight alterations, such as case changes, hyphenation, or the removal of Greek letters, are marked as “Auto-accepted Suggestion” and added to the Final List. Then, if a searched term still has not matched, it is compared with gene synonyms, and those with any potential matches (based on regular expression substring matching) are marked in the Final List and await the user to make a decision. These searched terms with ambiguous matches are selected one at a time from a dropdown, and their suggestions are listed along with other gene synonyms and descriptions. The most likely suggestion (based on the exact match of the searched term with a synonym) will be listed first. Lastly, those without any matches or those that are duplicate search terms are marked as “No Match” or “Duplicate Term” in the Final List, respectively. This functionality assists in the curation of a list of genes with officially recognized uniform identifiers.

The second task that the application assists with is the conversion of gene identifiers between formats, such as Ensembl IDs and official gene symbols recognized by the NCBI. Simply by entering a gene or list of genes into the text box, a list of curated genes is returned to the user as a table. In this way, gene IDs are converted without requiring the user to select the input type.

There are options to adjust the information included in the Final Table, and the user can save it as a .CSV file. Additionally, there are options to directly copy the full table, a list of the matched NCBI gene symbols, or other database IDs to the clipboard. Users may add genes to their curated Final List through multiple iterations of searches. Importantly, the total input and output lists will be the same order and length, to eliminate confusion in the case of large input lists. Finally, the application provides links to follow-up analyses such as ontology (PantherDB.org) that may be of interest now that the user has a curated and/or converted the list of uniform gene IDs.

### 2.3. Implementation

GeneToList was built as a web application in Python (3.8) using the Plotly Dash package (2.0.0), which provides a Python framework for web applications and relies on common JavaScript web frameworks: Flask (2.0.2), Plotly.js (5.5.0), and React.js. GeneToList is compatible with many modern browsers for desktop and mobile, including Google Chrome, Mozilla Firefox, Microsoft Edge, and Safari.

## 3. Results and Discussion

### 3.1. Gene Alias Disambiguation

To demonstrate GeneToList’s capacity for the disambiguation of obsolete or otherwise unofficial gene identifiers, we investigated a list of common inflammation-related genes retrieved from a recent publication [9]. These 10 genes were searched in GeneToList by the names by which they were referred (see “Searched Terms” in Table 1). GeneToList found four Exact Matches (Green in Table 1), one Auto-accepted Suggestion (Blue in Table 1), and five suggestions that required the user’s decision (Orange in Table 1). For example, *IL-8* had the suggestions: *CXCL8*, *CXCR1*, *CXCR2*, and *CXCR2P1*. Upon evaluation of the common synonyms listed, we determined that *CXCL8* was the best match. Additionally, because the algorithm in GeneToList found *CXCL8* to be the most likely choice, it was listed as the top suggestion. After similar decisions about the suggested selections for the remaining searched terms, we were left with a list of matched symbols (see “Matched Symbol” in Table 1).

### 3.2. Gene ID Conversion

Due to the disambiguation feature of GeneToList, it is able to serve the purpose of a gene ID converter, with better outcomes than other common ID converters. To demonstrate this, we used the same list of Searched Terms as in Table 1 and attempted their conversion to Entrez IDs using GeneToList, g:Convert, DAVID, and bioDBnet. These results are summarized in Table 2.

While GeneToList returned Entrez IDs for all searched terms, the other tools were only able to return 4 out of 10 queried. It is important to note that, when GeneToList was used first to disambiguate the terms (such as in Table 1) and then the “Matched Symbol” list was run through these other conversion tools, Entrez IDs were found for all (not shown). This is an example of the utility of disambiguation before a follow-up workflow.

## 4. Conclusions

The result of these efforts is a publicly available and free-to-use web application (GeneToList; https://www.genetolist.com/; accessed 1 June 2022) to assist biologists and biomedical scientists in navigating gene data. This tool assists in the disambiguation of gene IDs and was found to yield better results in ID conversion compared with the other common gene ID conversion tools. This is meant to aid in the uniformity of a list of genes before being used for any following analyses. Future efforts will be made to further expand the database of gene aliases and to continue optimizing the efficiency of the search algorithms.

## Figures and Tables

**Figure 1 biology-11-01113-f001:**
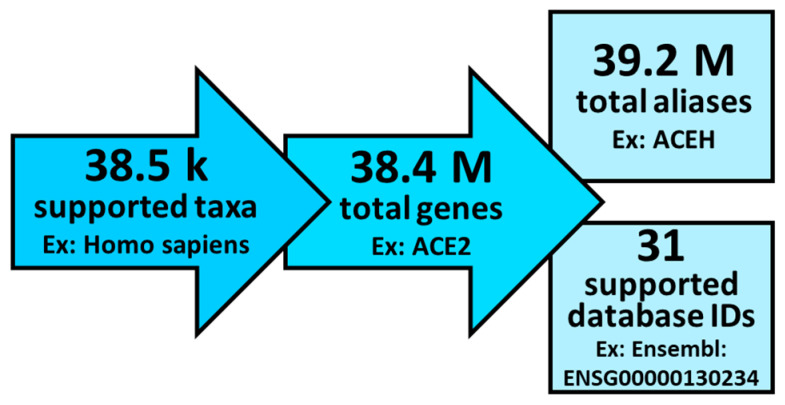
Overview of the gene information currently available to the GeneToList application.

**Table 1 biology-11-01113-t001:** Results of an example search of inflammation-related genes with GeneToList, demonstrating the disambiguation of gene IDs. Color corresponds to different match types where Green were exact matches between the searched term and the NCBI official symbol, Blue were auto-accepted suggestions where the searched term was similar to an official symbol and Orange were search terms which required alias disambiguation.

Searched Term	Match Type	Matched Symbol
*TGF-β*	**Suggestion Accepted**	*TGFB1*
*IL-8*	**Suggestion Accepted**	*CXCL8*
*MCP-1*	**Suggestion Accepted**	*CCL2*
*CRP*	**Exact Match**	*CRP*
*TNF-α*	**Suggestion Accepted**	*TNF*
*CXCR1*	**Exact Match**	*CXCR1*
*CXCR2*	**Exact Match**	*CXCR2*
*CCR2*	**Exact Match**	*CCR2*
*MYPT1*	**Suggestion Accepted**	*PPP1R12A*
*TGF-β1*	**Auto-accepted Suggestion**	*TGFB1*

**Table 2 biology-11-01113-t002:** Results of the gene ID conversion from GeneToList and other common conversion tools.

Searched Term	GeneToList	g:Convert	DAVID	bioDBnet
*TGF-β*	7040	-	-	-
*IL-8*	3576	-	-	-
*MCP-1*	6347	-	-	-
*CRP*	1401	1401	1401	1401
*TNF-α*	7124	-	-	-
*CXCR1*	3577	3577	3577	3577
*CXCR2*	3579	3579	3579	3579
*CCR2*	729,230	729,230	729,230	729,230
*MYPT1*	4659	-	-	-
*TGF-β1*	7040	-	-	-

## Data Availability

Not applicable.

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
