# Peer review of "GeneToList: A Web Application to Assist with Gene Identifiers for the Non-Bioinformatics-Savvy Scientist"

_biology, 2022, doi:10.3390/biology11081113_

Round 1

Reviewer 1 Report

This paper describes a useful web-based tool to disambiguate gene names and convert the corresponding identifiers. The tool addresses a long-standing problem in biomedical research, and seems to work well. The presentation is for the most part clear and to the point. 

One weakness of the paper is that it provides very little detail about how the name-matching algorithm works. What determines how "likely" a match is to be correct? Also, I was expecting the program to be able to deal with simple spelling errors (e.g., typing CXCK8 instead of CXCL8), but that's not the case. Is that because it would have been too computationally expensive, or because the authors didn't think it was a useful feature? Finally, how does the program handle cases in which a symbol is an "official" gene name, but also an alias for a different one?

In the Results section, Table 2 seems a bit pointless. It's obvious that the other tools listed, which rely on exact matches, are going to find results only for the genes that were spelled correctly in the first place. Maybe it could be replaced with some statistics about the contents of the database, e.g. average number of synonyms per gene, to better motivate the need for this tool?

Some comments about the interface: it would be useful if the external database IDs (e.g. ENSEMBL IDs) could be turned into hyperlinks, if the TaxID column showed the organism name in addition to the number, and if there was an option to download the results in tab-delimited form in addition to comma-delimited.  Finally, why does the "Suggested Matches" window only show three entries per page? 

Reviewer 2 Report

The paper presents a web application for disambiguation and search of gene IDs. The author clearly explain the need for such a tool and show some examples. The web app is up and working. I agree with the authors that this tool can be of great help to researchers.

Author Response

We thank the reviewer for their kind words and thoughtful consideration.